# Hybrid Inverted Index Is a Robust Accelerator for Dense Retrieval

**Peitian Zhang[1], Zheng Liu[2†], Shitao Xiao[2], Zhicheng Dou[1], Jing Yao[3]**

[1]Gaoling School of Artificial Intelligence, Renmin University of China
[2]Beijing Academy of Artificial Intelligence  [3]Microsoft Research Asia
{namespace.pt, zhengliu1026}@gmail.com

## Abstract

Inverted file structure is a common technique for accelerating dense retrieval. It clusters documents based on their embeddings; during searching, it probes nearby clusters w.r.t. an input query and only evaluates documents within them by subsequent codecs, thus avoiding the expensive cost of exhaustive traversal. However, the clustering is always lossy, which results in the miss of relevant documents in the probed clusters and hence degrades retrieval quality. In contrast, lexical matching, such as overlaps of salient terms, tends to be strong feature for identifying relevant documents. In this work, we present the Hybrid Inverted Index (HI$^2$), where the embedding clusters and salient terms work collaboratively to accelerate dense retrieval. To make best of both effectiveness and efficiency, we devise a cluster selector and a term selector, to construct compact inverted lists and efficiently searching through them. Moreover, we leverage simple unsupervised algorithms as well as end-to-end knowledge distillation to learn these two modules, with the latter further boosting the effectiveness. Based on comprehensive experiments on popular retrieval benchmarks, we verify that clusters and terms indeed complement each other, enabling HI$^2$ to achieve lossless retrieval quality with competitive efficiency across various index settings. Our code and checkpoint are publicly available at https://github.com/namespace-Pt/Adon/tree/HI2.

## 1 Introduction

Recently, dense retrieval has become the de-facto paradigm for high-quality first-stage text retrieval, serving as a fundamental component in various information access applications such as search engines (Zou et al., 2021), recommender systems (Zhao et al., 2022), and question answering systems (Karpukhin et al., 2020). Specifically, dense retrievers encode queries and documents into their latent *embeddings* in the semantic space using bi-encoders, and retrieve relevant documents based on embedding similarity. In practice, they rely on Approximate Nearest Neighbor (ANN) indexes to avoid expensive traversal of all document embeddings for each input query, a.k.a. the brute force search (Johnson et al., 2019).

There are numerous ANN options, e.g. the hashing based ones (Datar et al., 2004; Wang et al., 2018), the tree based ones (Bentley, 1975; Wang et al., 2014), the graph based ones (Wang et al., 2012; Malkov and Yashunin, 2018), and the vector quantization (VQ) based ones (Jégou et al., 2011a,b). Among all these alternatives, the VQ based indexes, exemplified by IVF-PQ, are particularly praised for their high running efficiency in terms of both query latency and space consumption, wherein the inverted file structure (IVF) is an indispensable component (Jégou et al., 2011a).

IVF partitions all document embeddings into disjoint clusters by KMeans. During searching, it finds nearby clusters to an input query and evaluates documents within these clusters by subsequent codecs (e.g. PQ). By increasing the number of clusters to scan, one may expect higher retrieval quality since the relevant document is more likely to be included, yet with higher query latency since there are more documents to evaluate (Jégou et al., 2011a). On top of the basic idea, recent studies improve the accuracy of IVF by grouping the cluster embeddings and skipping the least promising groups (Baranchuk et al., 2018), creating duplicated records for boundary embeddings (Chen et al., 2021), and end-to-end learning the cluster assignments by knowledge distillation (Xiao et al., 2022a). Despite their improvements, IVF still exhibits limited retrieval quality, especially when high efficiency is needed. This is because the clustering is too lossy to include relevant documents in a few close clusters to the query. What's worse, it is not cost-effective to probe more clusters, which

---

[†]Corresponding Author.

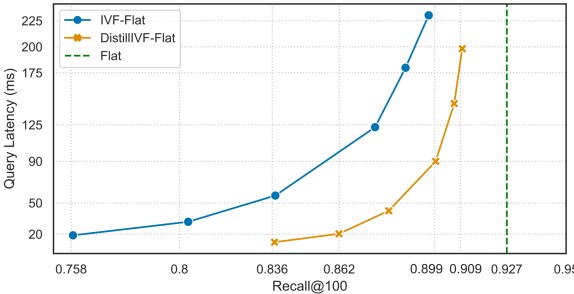

Figure 1: Recall-latency trade-off example for existing IVF methods on MSMARCO Passage. Better indexes should locate at the lower right corner.

sacrifices much more efficiency for minor effectiveness improvements. To better illustrate the above points, we take a concrete example.

**Example 1** *In Figure 1, we showcase the recall-latency trade-off derived from changing the number of clusters to visit in the basic IVF and the distilled IVF (the best IVF method so far (Xiao et al., 2022a)). We use the "Flat" codec that reranks the candidate documents in brute force. As such, any retrieval loss against brute force search (denoted as "Flat") is due to the failure of IVF.*

*Two critical facts can be observed. First, despite improvements from end-to-end distillation, both IVF methods suffer from much poorer retrieval quality at low query latency. With 20ms latency, IVF-Flat and DistillIVF-Flat achieve recall of 0.758 and 0.862, both of which lag far behind 0.927 from brute force search. Second, probing more clusters marginally improves recall but significantly increases latency. To promote recall from 0.899 to 0.909, DistillIVF-Flat doubles the query latency (from 90ms to around 200ms). Consequently, there is plenty of room to optimize IVF to achieve lossless retrieval quality with high efficiency.*

In contrast to cluster proximity, extensive research has demonstrated that lexical matching, e.g. overlaps of salient terms between queries and documents, tend to be strong features for identifying relevant documents (Robertson and Zaragoza, 2009; Lin and Ma, 2021; Formal et al., 2021). Moreover, complementary effect has been observed from combining lexical and semantic matching in hybrid retrieval systems (Kuzi et al., 2020; Gao et al., 2020; Shen et al., 2022; Zhang et al., 2023).

In this work, we explore the potential of unifying embedding clusters and salient terms in a **H**ybrid **I**nverted **I**ndex (HI$^2$) for the acceleration of dense retrieval. Specifically, each document reference

is indexed in inverted lists of two types of entries: embedding clusters and salient terms. When searching, the input query is dispatched to both types of inverted lists. Documents within them are merged and evaluated by subsequent codecs.

For effectiveness, HI$^2$ needs to include relevant documents in the dispatched inverted lists; For efficiency, HI$^2$ requires these inverted lists to be small enough to avoid significant overhead during post-hoc evaluation. Both of them call for constructing compact inverted lists and efficiently searching through them. To this end, we devise a cluster selector and a term selector, which accurately and efficiently pick out only a few clusters and terms for indexing and searching, respectively.

As for the implementation of the cluster and term selector, we show simple unsupervised algorithms, i.e. KMeans and BM25, work surprisingly well, whereby HI$^2$ already substantially outperforms previous IVF methods with competitive efficiency. Moreover, we propose to leverage neural networks for realization and end-to-end learning by a knowledge distillation objective. This approach further boosts the retrieval quality, enabling HI$^2$ to remarkably and consistently surpass other ANN indexes on popular retrieval benchmarks, i.e. MS-MARCO (Nguyen et al., 2016) and Natual Questions (Kwiatkowski et al., 2019).

Our contributions are summarized as follows:

- We propose the Hybrid Inverted Index, which combines embedding clusters and salient terms for accelerating dense retrieval.

- We devise tailored techniques, i.e. the cluster selector, the term selector, and the joint optimization, to guarantee the effectiveness and efficiency of HI$^2$.

- We evaluate HI$^2$ with extensive experiments and verify its robust advantage across implementation variations, indexing/searching configurations, and embedding models.

## 2 Related Works

. ● **Dense Retrieval**. In the last four years, the rapid development of pre-trained language models, e.g. BERT (Devlin et al., 2019), has significantly pushed forward the progress of dense retrieval, making it increasingly popular for high-quality first-stage retrieval (Zhao et al., 2022; Zhu et al., 2023). Dense retrievers encode queries and

documents into dense vectors (i.e. *embeddings*) in the same latent space, where the semantic relevance is measured by embedding similarity. Recent studies further enhance their retrieval quality by retrieval-oriented pre-training (Wang et al., 2022; Xiao et al., 2022b; Gao and Callan, 2022), delicate negative sampling (Xiong et al., 2021; Qu et al., 2021; Zhan et al., 2021b), and knowledge distillation from more powerful rankers (Zhang et al., 2022; Lu et al., 2022; Qu et al., 2021).

● **ANNs Indexes**. In practice, relevant documents usually need to be retrieved from a massive collection. Consequently, dense retrieval must rely on Approximate Nearest Neighbor (ANN) indexes to avoid the expensive brute force search. The ANNs indexes can be realized via different strategies: 1) the hashing based ones (Datar et al., 2004; Weiss et al., 2008; Wang et al., 2018); 2) the tree based ones (Bentley, 1975; Wang et al., 2014; Muja and Lowe, 2014); 3) the graph based ones (Dong et al., 2011; Wang et al., 2012; Malkov and Yashunin, 2018); 4) the vector quantization (VQ) based ones (Ge et al., 2014; Jégou et al., 2011a,b; Baranchuk et al., 2018). Among these options, the VQ based indexes are particularly preferred for massive-scale retrieval owing to their high efficiency in terms of both query latency and space consumption (Johnson et al., 2019).

● **VQ Index Optimization**. Despite the competitive efficiency, VQ-based indexes are prone to limited retrieval quality when low latency is desired. In recent years, continuous efforts have been dedicated to alleviating this problem, which can be categorized into two threads. One thread is to design advanced heuristics for clustering and evaluation. For example, (Jégou et al., 2011b) and (Baranchuk et al., 2018) add another refinement stage over the quantized embeddings and skip less promising clusters according to tailored heuristics. (Chen et al., 2021) create duplicated reference for boundary embeddings to improve recall with high efficiency. The other research thread optimizes the VQ index towards retrieval quality with cross-entropy loss instead of minimizing the reconstruction loss. For example, (Zhan et al., 2021a) and (Xiao et al., 2021) jointly learns the query encoder and the product quantizer by contrastive learning. (Xiao et al., 2022a) further improves the accuracy by leveraging knowledge distillation for joint optimization. However, all these methods stick to conventional IVF to organize the search space, which is subop-

timal as shown in Example 1. In this work, our proposed Hybrid Inverted Index support efficient identification of relevant documents through both semantic and lexical matching. Note that our work is orthogonal to those about efficient inverted index access (Broder et al., 2003; Mallia et al., 2022) and hence can be combined for further acceleration.

● **Hybrid Retrieval**. Recently, there have been emergent recipes for the union of semantic (dense) and lexical (sparse) features. Some of them are direct ensembles of dense and sparse retrievers (Ma et al., 2021; Kuzi et al., 2020); Others use enhanced optimization objectives, e.g. adversarial hard negatives and distillation, to jointly learn from semantic/lexical features (Gao et al., 2020; Shen et al., 2022; Zhang et al., 2023). However, they all rely on separate sparse and dense indexes to work, and interpolate scores from the two indexes. Different from them, HI$^2$ combines semantic and lexical features at index level, and estimates scores universally by specific codecs. Meanwhile, HI$^2$ may benefit from enhanced optimization methods in these methods, which we leave for future work.

## 3 Preliminaries

### 3.1 Dense Retrieval

Given a collection of documents $\mathcal{D} = \{D_i\}_{i=1}^{|\mathcal{D}|}$, dense retrieval aims to retrieve the top $R$ relevant documents from $\mathcal{D}$ in response to an input query $Q$. Specifically, each document $D \in \mathcal{D}$ and query $Q$ is encoded into its embedding $\boldsymbol{e}_D, \boldsymbol{e}_Q \in \mathbb{R}^h$, by a document encoder and a query encoder, respectively. Next, the relevance is measured by the inner product between them, whereby the top $R$ ranked documents are returned.

$$\text{result} = \underset{D}{\text{top-}R} \left\{ \langle \boldsymbol{e}_Q, \boldsymbol{e}_D \rangle \mid D \in \mathcal{D} \right\}, \quad (1)$$

where $\langle \cdot, \cdot \rangle$ denotes inner product.

In reality, it is impractical to evaluate every document (computing $\langle \boldsymbol{e}_Q, \boldsymbol{e}_D \rangle$) for each input query (i.e. the *brute force search*), which results in exceedingly high latency and resource consumption. Instead, ANN indexes are used to avoid exhaustively scanning all documents and accelerate the relevance measurement by approximation.

### 3.2 Inverted File Structure and Product Quantization

Among all alternative ANN indexes, the Vector Quantization (VQ) based ones are particularly popular for massive-scale retrieval. They consist of

two basic modules: the inverted file structure (IVF) and the product quantization (PQ).

To avoid exhaustive search, IVF partitions all documents into disjoint clusters $\boldsymbol{C} = \{C_i\}_{i=1}^{L}$ by KMeans, where each cluster is associated with an embedding $\boldsymbol{e}_{C_i} \in \mathbb{R}^h$. For the query $Q$, documents within the closest $K^C$ clusters are evaluated by the subsequent codec (PQ by default):

$$\text{result} = \underset{D}{\text{top-}}R\left\{\text{PQ}(\text{Q}, \text{D}) \mid D \in \mathcal{A}(Q)\right\},$$
$$\mathcal{A}(Q) = \bigcup_{C_i} \text{top-}K^C\left\{\langle \boldsymbol{e}_Q, \boldsymbol{e}_{C_i}\rangle \mid C_i \in \boldsymbol{C}\right\}. \quad (2)$$

To accelerate relevance estimation, PQ compresses the document embedding into discrete integer codes according to a codebook $\boldsymbol{v} \in \mathbb{R}^{m \times k \times h/m}$. It splits $\boldsymbol{e}_D$ into $m$ fragments $\{\boldsymbol{e}_D^j\}_{j=1}^{m}$, then quantizes each fragment to the closest codeword in $\boldsymbol{v}$:

$$\hat{\boldsymbol{e}}_D^j = \boldsymbol{v}_{j,\theta_j}, \quad \theta_j = \underset{i}{\text{argmin}} ||\boldsymbol{e}_D^j - \boldsymbol{v}_{j,i}||_2^2. \quad (3)$$

Therefore, only the global codebook $\boldsymbol{v}$ and the codeword assignment $\theta_*$ need to be stored, which is *much smaller* than the full-precision embedding. Finally, the relevance is evaluated by:

$$\text{PQ}(Q, D) = \sum_{j=1}^{m} \langle \boldsymbol{e}_Q^j, \hat{\boldsymbol{e}}_D^j \rangle, \quad (4)$$

where $\boldsymbol{e}_Q^j$ is the query embedding fragment. Since the inner product between $\boldsymbol{e}_Q^j$ and any codeword $\boldsymbol{v}_{j,*}$ can be cached once computed, the relevance estimation approximated by PQ is *much faster*.

By increasing the number of clusters to scan ($K^C$), higher retrieval quality can be achieved because the relevant document is more likely to be included in $\mathcal{A}(Q)$. Yet, the latency is increased at the same time as more documents need to be evaluated. Conventional IVF falls short in including the relevant document given a small $K^C$, meanwhile, it needs to sacrifice a lot of efficiency for minor retrieval quality improvement. In this work, we propose an alternative to alleviate these problems.

## 4  Hybrid Inverted Index

The framework of the Hybrid Inverted Index (HI²) is shown in Figure 2. HI² organizes the search space with two types of inverted lists: embedding clusters ($\boldsymbol{C} = \{C_i\}_{i=1}^{L}$) and salient terms ($\boldsymbol{T} = \{T_v \mid v \in \mathcal{V}\}$ where $\mathcal{V}$ is the term vocabulary). Each document reference is stored in the inverted

lists of 1 cluster and $K_1^T$ terms. When searching, the input query $Q$ is dispatched to the inverted lists of $K^C$ clusters and $K_2^T$ terms. Documents within them are merged and evaluated by PQ. Formally,

$$\text{result} = \underset{D}{\text{top-}}R\left\{\text{PQ}(\text{Q}, \text{D}) \mid D \in \mathcal{A}(Q)\right\},$$
$$\mathcal{A}(Q) = \mathcal{A}^C(Q) \cup \mathcal{A}^T(Q). \quad (5)$$

To determine which clusters/terms to use for indexing the document and dispatching the query, HI² employs two modules: a cluster selector and a term selector. They can be implemented with simple unsupervised algorithms (resulting in HI²$_{\text{unsup}}$) and neural networks (resulting in HI²$_{\text{sup}}$).[1] This flexible implementation scheme injects high practicability into HI². In the following, we elaborate on the two modules (§4.1 and §4.2) and the supervised optimization for their neural network implementation (§4.3).

### 4.1  Cluster Selector

This module selects 1 cluster for indexing the document and $K^C$ clusters for dispatching the query to. Specifically, it associates each cluster $C_i$ with an embedding $\boldsymbol{e}_{C_i} \in \mathbb{R}^h$. Then scores each cluster with the inner product between the document embedding $\boldsymbol{e}_D$ or the query embedding $\boldsymbol{e}_Q$: $\langle \boldsymbol{e}_*, \boldsymbol{e}_{C_i}\rangle$. The document is indexed to the cluster with the highest score. When searching, the query is dispatched to the top $K^C$ clusters:

$$\mathcal{A}^C(Q) = \bigcup_{C_i} \text{top-}K^C\left\{\langle \boldsymbol{e}_Q, \boldsymbol{e}_{C_i}\rangle \mid C_i \in \boldsymbol{C}\right\}. \quad (6)$$

For HI²$_{\text{unsup}}$, the cluster embeddings $\{\boldsymbol{e}_{C_i}\}_{i=1}^{L}$ are produced by KMeans over all document embeddings. For HI²$_{\text{sup}}$, they are initialized with KMeans and optimized on-the-fly by the objective in §4.3.

### 4.2  Term Selector

This module selects $K_1^T$ terms for indexing the document and $K_2^T$ terms for dispatching the query. There are two concerns for designing the term selector: 1) the selected terms must be representative w.r.t. the input, hence the lexical matching between the query and the document can be effectively captured; 2) the term selection for the query must be efficient enough to avoid excess online overhead.

Therefore, for the document $D$, the term selector first tokenizes it to $\{d_i\}_{i=1}^{|D|}$ where $d_i \in \mathcal{V}$, then it

---

[1]We use HI² to denote both HI²$_{\text{unsup}}$ and HI²$_{\text{sup}}$ henceforth.

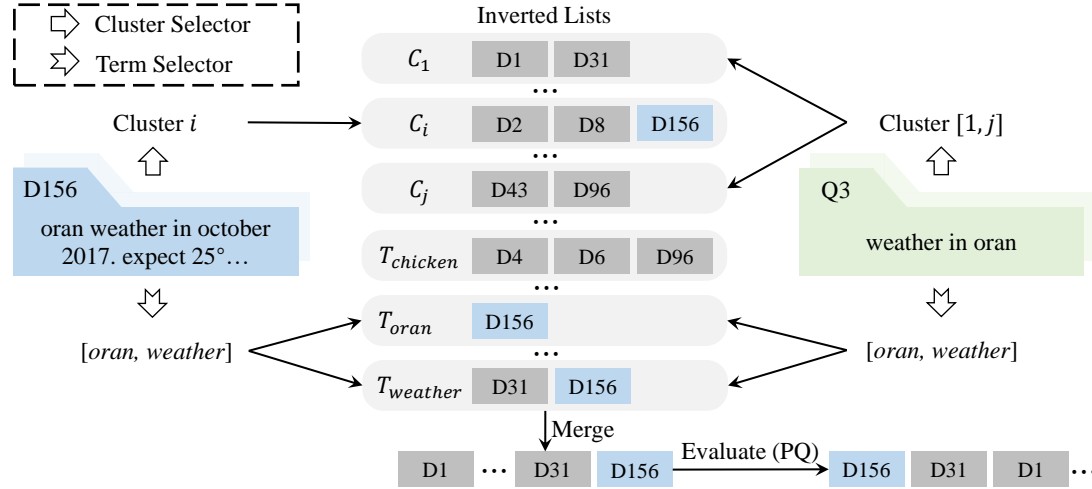

Figure 2: The framework of the Hybrid Inverted Index (HI²). Each document reference is indexed in inverted lists of two types of entries: embedding clusters and salient terms. When searching, the input query is dispatched to both types of inverted lists. Documents within them are merged and evaluated by the subsequent codec (PQ).

estimates the score of each unique term $v \in \mathcal{V}$ in $D$ with BM25 (HI²$_{\text{unsup}}$) or BERT (HI²$_{\text{sup}}$). Formally,

$$s_v = \begin{cases} \frac{(\alpha+1)\times\text{IDF}(v)\times\text{TF}(v,D)}{\text{TF}(v,D)+\alpha\times(1-\beta+\frac{\beta|D|}{\text{avgdl}})} & \exists d_i = v \wedge \text{HI}^2_{\text{unsup}}, \\ \max_{d_i=v}(f(\text{BERT}(D)[i])) & \exists d_i = v \wedge \text{HI}^2_{\text{sup}}, \\ 0 & \nexists d_i = v. \end{cases}$$
$$(7)$$

$\alpha, \beta$ are hyper parameters, avgdl is the average document length, $f(\cdot)$ is two-layer MLP of $\mathbb{R}^h \to \mathbb{R}^1$ with ReLU activation, and BERT denotes encoding by BERT model (Devlin et al., 2019). As such, the top $K_1^T$ scored terms are used for indexing the document. Besides, the average score of each term across all documents ($\bar{s}_v$) is stored.

The query $Q$ is tokenized to $\{q_i\}_{i=1}^{|Q|}$ likewise, while it is not processed with any complex computations to save online cost. For short queries, all its constituting terms are selected; for long queries, the terms with top $K_2^T$ average score are selected:

$$\mathcal{A}^T(Q) = \begin{cases} \bigcup\{T_{q_i} \mid q_i \in Q\} & |Q| \leq K_2^T, \\ \bigcup\{T_{q_i} \mid q_i \in Q'\} & \text{otherwise}. \end{cases}$$
$$Q' = \underset{q_i}{\text{top-}K_2^T}\{\bar{s}_{q_i} \mid q_i \in Q\} \quad (8)$$

### 4.3 Joint Optimization

HI²$_{\text{sup}}$ involves learning cluster embeddings in the cluster selector, the MLP, and BERT in the term selector. We propose a knowledge distillation objective for jointly training these parameters towards retrieval quality. Concretely, we sample a subset of documents for the query ($\boldsymbol{D} \subseteq \mathcal{D}$), then employ

a powerful teacher $\boldsymbol{\Theta}$ to produce accurate estimations of their relevance. Finally, we enforce the cluster selector and the term selector to produce similar estimations by KL divergence:

$$\mathcal{L}^{\text{Distill}}(Q) = \text{KL}(\boldsymbol{\Theta}(Q, \boldsymbol{D}) \parallel \text{CS}(Q, \boldsymbol{D})) \\ + \text{KL}(\boldsymbol{\Theta}(Q, \boldsymbol{D}) \parallel \text{TS}(Q, \boldsymbol{D})). \quad (9)$$

Following (Xiao et al., 2022a), we simply choose off-the-shelf embeddings as teachers. Denote softmax operator as sm, the teacher estimations are:

$$\boldsymbol{\Theta}(Q, \boldsymbol{D}) = \text{sm}\{\langle \boldsymbol{e}_Q, \boldsymbol{e}_D \rangle \mid D \in \boldsymbol{D}\}. \quad (10)$$

The cluster selector estimates relevance by query-cluster embedding similarity:

$$\text{CS}(Q, \boldsymbol{D}) = \text{sm}\{\langle \boldsymbol{e}_Q, \boldsymbol{e}_{C_{\phi(D)}} \rangle \mid D \in \boldsymbol{D}\}, \quad (11)$$

where $\phi(D)$ is the cluster index of the document.

The term selector estimates relevance by term-score vector similarity:

$$\text{TS}(Q, \boldsymbol{D}) = \text{sm}\{\langle \boldsymbol{s}_Q, \boldsymbol{s}_D \rangle \mid D \in \boldsymbol{D}\}, \quad (12)$$

where $\boldsymbol{s}_Q$ and $\boldsymbol{s}_D$ are the score vector over the vocabulary derived from Eq 7. Note that here both queries and documents are processed the same way.

Additionally, since the document cluster assignment $\phi(D)$ is fixed, we add a commitment loss to the final loss to keep the document embedding close to its associated cluster, which is a common practice for learning quantization (van den Oord

et al., 2017):

$$\mathcal{L}^{\text{Commit}}(Q) = \sum_{D \in \boldsymbol{D}} \log \frac{\exp(\langle \boldsymbol{e}_D, \boldsymbol{e}_{C_{\phi(D)}} \rangle)}{\sum_{C' \in \boldsymbol{C}} \exp(\langle \boldsymbol{e}_D, \boldsymbol{e}_{C'} \rangle)},$$

$$\mathcal{L} = \sum_Q \mathcal{L}^{\text{Distill}}(Q) + \mathcal{L}^{\text{Commit}}(Q). \quad (13)$$

## 5 Experiments

In this section, we first introduce our experimental settings, then carry out extensive experiments to investigate the following research questions (RQ):
**RQ1:** *How are the effectiveness and efficiency of $HI^2$ compared with other baselines methods?*
**RQ2:** *Do clusters and terms complement each other for identifying relevant documents?*
**RQ3:** *How is the robustness of $HI^2$ across different embedding models?*

### 5.1 Experimental Settings

• **Datasets.** We use two popular benchmark datasets. 1) MS MARCO (Nguyen et al., 2016). We use the passage track, including 502,939 training queries and 6,980 evaluation queries (dev set); the corpus size is 8,841,823. 2) Natural Questions (Kwiatkowski et al., 2019). We follow the split of DPR (Karpukhin et al., 2020), resulting in 79,168 training queries and 3,610 testing queries. The corpus size is 21,015,324.

• **Metrics.** For evaluating retrieval quality, we leverage MRR@K (M@K) and Recall@K (R@K). For evaluating retrieval efficiency, we compute the average query latency (QL) and the overall index size (IS). Our evaluations are based on the same batch size, thread number, and toolkit (Faiss (Johnson et al., 2019) for ANNs and Pyserini (Lin et al., 2021) for sparse models). Note that the latency of $HI^2$ is on par with that of IVF-OPQ given the same number of candidates to evaluate, because the term selector dispatches the query with simple heuristics that introduces very little overhead.

• **Baselines.** We compare with three types of retrieval models: 1) Sparse retrievers, including BM25 (Robertson and Zaragoza, 2009), DocT5 (Cheriton, 2019), DeepCT (Dai and Callan, 2019), UniCOIL (Lin and Ma, 2021), and Distil-SPLADE (Formal et al., 2021); 2) Dense retrievers with brute force search (denoted as Flat), including DPR (Karpukhin et al., 2020), ANCE (Xiong et al., 2021), CoCondenser (Gao and Callan, 2022), AR2 (Zhang et al., 2022), and RetroMAE (Xiao

et al., 2022b). 3) Dense retrievers with ANN indexes (denoted as ANNs), including IVF-PQ (Jégou et al., 2011a), IVF-OPQ (Ge et al., 2014), IVF-JPQ (Zhan et al., 2021a), Distill-VQ (Xiao et al., 2022a), and HNSW (Malkov and Yashunin, 2018) (we set the edges to 32 and efSearch to 500). We use **RetroMAE** and **AR2** as the embedding model on MSMARCO and NQ, respectively, due to their superior performance with brute force search. More details are in the Appendix.

• **Implementation Details.** For all methods involving clustering, we set the number of clusters $L$ to 10000 and the number of probing clusters when searching to 100 (except $HI^2$). For all methods involving PQ, we set the number of fragments $m$ to 96, the number of sub-clusters $k$ to 256. For $HI^2$, we use the BERT's vocabulary (Devlin et al., 2019) as the term vocabulary $\mathcal{V}$, resulting in 30522 unique terms in total. $K_2^T$ is always set to 32 for both $HI^2_{\text{unsup}}$ and $HI^2_{\text{sup}}$.

For $HI^2_{\text{unsup}}$, we use KMeans over all document embeddings to produce cluster embeddings $\{\boldsymbol{e}_{C_i}\}_{i=1}^L$, BM25 to produce term scores $s_v$ with $\alpha = 0.82, \beta = 0.68$, and OPQ (Ge et al., 2014) as the evaluation codec, all of which are unsupervised algorithms. $K^C$ is set to 25, $K_1^T$ is set to 15. For $HI^2_{\text{sup}}$, we initialize cluster embeddings with KMeans and optimize them afterward. Note the cluster assignment $\phi(D)$ is fixed once initialized. We use bert-base-uncased for the term selector. The passage is tokenized to 128 tokens before encoding. We employ the distilled OPQ (Xiao et al., 2022a) as the evaluation codec. $K^C$ is set to 30, $K_1^T$ is set to 3. More details are in the appendix. For reproducibility, we release our source code and model checkpoints at `https://anonymous.4open.science/r/HI2/`.

### 5.2 Main Analysis (RQ1)

We report the overall evaluation results in Table 1.

On the one hand, our hybrid inverted index demonstrate superlative **effectiveness** over baseline ANN indexes. Specifically, $HI^2_{\text{unsup}}$, which solely relies on unsupervised algorithms, improves the Recall@100 of IVF-OPQ (the basic unsupervised VQ index) by 14%, and improves that of Distill-VQ (the strongest supervised VQ index in literature) by 8%. It even triumphs the powerful HNSW index by 3 absolute points in Recall@100, which is a more valuable metric for first-stage retrieval than MRR. Moreover, the neural network imple-

| Type | Method | MS MARCO | | | | | Natual Questions | | | | |
|---|---|---|---|---|---|---|---|---|---|---|---|
| | | M@10 | R@100 | R@1000 | QL (ms) | IS (G) | R@5 | R@20 | R@100 | QL (ms) | IS (G) |
| Sparse Retrievers | BM25 | 0.187 | 0.592 | 0.670 | 12 | **0.6** | 0.490 | 0.639 | 0.788 | 41 | 2.3 |
| | DocT5 | 0.277 | – | 0.947 | 17 | 1.0 | – | – | – | – | – |
| | DeepCT | 0.243 | – | 0.913 | 12 | 0.7 | – | – | – | – | – |
| | UniCOIL | 0.330 | 0.823 | 0.932 | 124 | 1.0 | 0.638 | 0.774 | 0.861 | 480 | 3.4 |
| | DistilSPLADE | 0.366 | 0.896 | 0.978 | 455 | 2.4 | – | – | – | – | – |
| Dense Retrievers (Flat) | DPR | 0.317 | 0.857 | 0.959 | 1751 | 26 | – | 0.784 | 0.853 | 4785 | 60 |
| | ANCE | 0.346 | 0.873 | 0.964 | 1751 | 26 | – | 0.819 | 0.875 | 4785 | 60 |
| | CoCondenser | 0.382 | – | 0.984 | 1751 | 26 | 0.758 | 0.843 | 0.890 | 4785 | 60 |
| | AR2 | 0.395 | 0.919 | 0.984 | 1751 | 26 | **0.779** | **0.861** | **0.908** | 4785 | 60 |
| | RetroMAE | **0.416** | **0.927** | **0.988** | 1751 | 26 | – | 0.844 | 0.894 | 4785 | 60 |
| Dense Retrievers (ANNs) | IVF-PQ | 0.292* | 0.763* | 0.849* | 13 | 0.9 | 0.545* | 0.696* | 0.799* | 28 | **2.1** |
| | IVF-OPQ | 0.346* | 0.796* | 0.853* | 13 | 0.9 | 0.687* | 0.775* | 0.836* | 28 | **2.1** |
| | IVF-JPQ | 0.348* | 0.799* | 0.854* | 13 | 0.9 | 0.689* | 0.776* | 0.836* | 28 | **2.1** |
| | Distill-VQ | 0.358* | 0.843* | 0.914* | 10 | 0.9 | 0.705* | 0.799* | 0.860* | 16 | **2.1** |
| | HNSW | 0.400 | 0.887* | 0.944* | **6** | 28 | 0.778 | 0.857 | 0.898 | **13** | 66 |
| Ours | $\text{HI}^2_{\text{unsup}}$ | 0.380 | 0.900 | 0.966 | 9 | 3.0 | 0.767 | 0.853 | 0.896 | 15 | 7.1 |
| | $\text{HI}^2_{\text{sup}}$ | 0.401 | 0.916 | 0.976 | 8 | 3.0 | **0.779** | **0.861** | 0.906 | 15 | 7.1 |

Table 1: Overall evaluation results. Statistically significant results within ANNs group compared with $\text{HI}^2_{\text{sup}}$ (paired t-test with $p < 0.05$) are decorated with *. Best results are bold. Second best results are underlined. QL means the average query latency. IS means the index size.

mentation and the end-to-end knowledge distillation further unleash its potential, as $\text{HI}^2_{\text{sup}}$ further amplify the margins over ANN baselines. Remarkably, $\text{HI}^2_{\text{sup}}$ achieves on par retrieval quality with its brute-force-search teacher (RetroMAE on MS-MARCO and AR2 on NQ), and surpasses a lot of well-established sparse and dense retrievers.

On the other hand, the **efficiency** of our hybrid inverted index is also satisfactory. Its query latency is the second lowest on both datasets, accelerating the brute force search (Flat) by hundreds of times and only slightly falling behind that of HNSW. Notably, the latency is even lower than VQ-based indexes, because $\text{HI}^2$ needs to evaluate fewer candidate documents. Besides, $\text{HI}^2$ possesses a moderate index size, which is bigger than VQ baselines since more document references need to be stored, while much smaller than Flat or HNSW since it does not need to store full-precision embeddings.

As such, we have showcased the outstanding effectiveness and efficiency of $\text{HI}^2$ under one specific setting. Next, we are interested in the effectiveness-efficiency trade-off of $\text{HI}^2$ and ANN baselines (we exclude IVF-PQ and IVF-JPQ, the former is too weak and the latter is similar to IVF-OPQ). Specifically, for VQ indexes, we change the number of clusters to visit; for HNSW, we change the number of neighbors to visit; for $\text{HI}^2$, we change the number of terms to index ($K_1^T$) and the number of clusters to dispatch ($K^C$). Since the index size is static, we measure recall@100 as effectiveness and average query latency as efficiency. The resulted trade-off curves are reported in Figure 3.

From the figure, $\text{HI}^2_{\text{unsup}}$ performs on par with the powerful HNSW across various index settings, as their recall are almost identical given the same latency. Both of them significantly outperform VQ baselines. Besides, $\text{HI}^2_{\text{sup}}$ brings substantial improvement over $\text{HI}^2_{\text{unsup}}$ and HNSW, achieving higher recall with lower latency. Meanwhile, it efficiently approaches the brute-force-search effectiveness. In contrast, VQ baselines need to largely increase the latency to marginally improve the recall, yet lagging far behind brute force search.

Based on the above analysis, we answer RQ1: *$\text{HI}^2$ achieves lossless retrieval quality against brute force search, with low query latency and small index size, significantly and consistently outperforming baseline ANN indexes and retaining the advantages across indexing/searching configurations.*

### 5.3 Ablation Analysis (RQ2)

To answer RQ2, we study the individual contribution from embedding clusters and salient terms. Specifically, we disable the inverted lists corresponding to terms and clusters, respectively, denoted as *w.o. Term* and *w.o. Clus*. Other configurations are kept the same. We plot their recall-latency trade-off curves in Figure 4.

Two critical facts can be observed. First, salient terms tend to be better features for organizing the search space than embedding clusters, as the *w.o. Clus* variants significantly and consistently outperform *w.o. Term* ones. Thus, our claim that embedding clusters alone falls short in effective identification of relevant documents is well justified. Second,

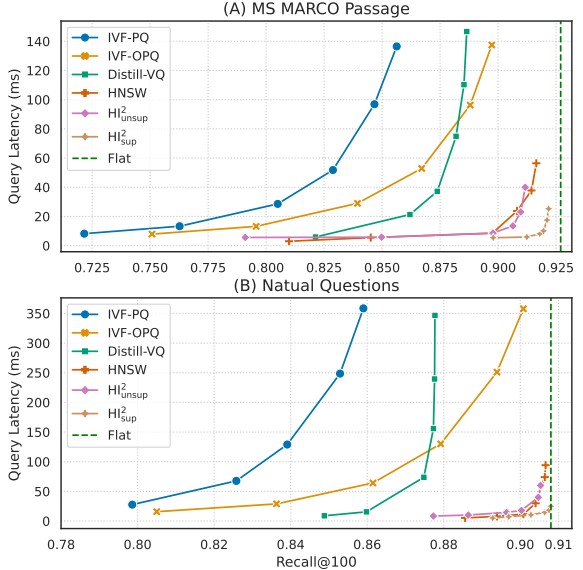

Figure 3: Effectiveness-efficiency trade-off of different methods on MS MARCO (A) and Natual Questions (B).

| Emb. | ANN Index | MS MARCO | | | NQ | | |
|---|---|---|---|---|---|---|---|
| | | R@100 | QL | IS | R@100 | QL | IS |
| RetroMAE | Flat | **0.927** | 1751 | 26 | **0.894** | 4785 | 60 |
| | HNSW | 0.887 | **6** | 28 | 0.873 | **12** | 66 |
| | IVF-OPQ | 0.796 | 13 | **0.9** | 0.843 | 19 | **2.1** |
| | Distill-VQ | 0.843 | 10 | **0.9** | 0.870 | 19 | **2.1** |
| | $HI^2_{unsup}$ | 0.900 | 9 | 3.0 | 0.880 | 15 | 7.1 |
| | $HI^2_{sup}$ | 0.916 | 8 | 3.0 | 0.885 | 15 | 7.1 |
| AR2 | Flat | **0.919** | 1751 | 26 | **0.908** | 4785 | 60 |
| | HNSW | 0.904 | 7 | 28 | 0.898 | **13** | 66 |
| | IVF-OPQ | 0.846 | 11 | **0.9** | 0.836 | 28 | **2.1** |
| | Distill-VQ | 0.858 | 10 | **0.9** | 0.860 | 16 | **2.1** |
| | $HI^2_{unsup}$ | 0.899 | 9 | 3.0 | 0.896 | 15 | 7.1 |
| | $HI^2_{sup}$ | 0.909 | 8 | 3.0 | 0.906 | 15 | 7.1 |

Table 2: Evaluation of $HI^2$ and strong ANN baselines with different embedding models (Emb.).

salient terms and embedding clusters indeed complement each other, as $HI^2_{unsup}$ and $HI^2_{sup}$ beats their "homogeneous" variants in terms of both effectiveness and efficiency. Therefore, we answer RQ2: *Embedding clusters and salient terms complement each other for more effective and efficient identification of relevant documents.*

## 5.4 Robustness Analysis (RQ3)

In Figure 3, we have shown the robust advantage of $HI^2$ across different index configurations. For practical usage, it is important to evaluate the robustness of $HI^2$ given different embedding models.

In Table 2, we report the performance of $HI^2$ and selected strong baselines with RetroMAE and AR2 as the embedding model. We can notice that $HI^2_{sup}$ always achieves the best recall among all ANN indexes, which is very close to that of brute force search. Besides, $HI^2_{unsup}$ performs on par with

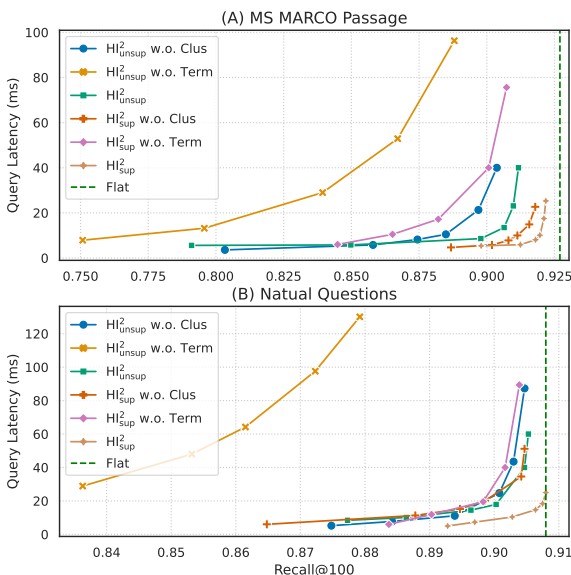

Figure 4: Effectiveness-efficiency trade-off of $HI^2$ variants on MS MARCO (A) and Natual Questions (B).

the strong HNSW index, which uses full-precision embeddings for evaluation. As for efficiency, we observe $HI^2$ resides in a sweet spot with the second lowest query latency and relatively small index size, which is substantially smaller than Flat and HNSW but slightly bigger than VQ baselines.

Additionally, the performance of ANN baselines is unstable given different embedding models: higher retrieval quality with brute-force searching does not result in higher retrieval quality with ANN acceleration. For example, the recall of AR2 Flat is inferior to that of RetroMAE Flat on MS MARCO. However, this trend reverses when ANN baselines are applied, i.e. AR2 IVF-OPQ is better than Retro-MAE IVF-OPQ. By comparison, the performance of $HI^2$ is stable: higher brute-force-search effectiveness corresponds to higher effectiveness of $HI^2$ regardless of the embedding model.

In summary, we answer RQ3: *$HI^2$ enjoys high robustness and stability across different embedding models, consistently surpassing strong ANN baselines with competitive efficiency and aligning well with the brute force search.*

## 6 Conclusion

In this work, we propose the hybrid inverted index, which reformulates conventional IVF by unifying both embedding clusters and salient terms to accelerate dense retrieval. We devise tailored techniques for cluster selection, term selection, and joint optimization. With comprehensive experiments, we verify the effectiveness and efficiency

of HI$^2$, which consistently outperforms strong ANN baselines across implementation variations, indexing/searching configurations, and embedding models. Moreover, we demonstrate that embedding clusters and salient terms are complementary to each other for identifying relevant documents, which may inspire further research towards the combination of semantic and lexical features.

## Acknowledgement

This work was supported by the National Natural Science Foundation of China No. 62272467 and Public Computing Cloud, Renmin University of China. The work was partially done at the Engineering Research Center of Next-Generation Intelligent Search and Recommendation, MOE.

## Limitations

Despite the satisfactory performance of the hybrid inverted index, it has more hyper parameters than conventional IVF hence may require more effort to tune them for ideal performance. Moreover, the searching of clusters and terms is currently independent; whilst we believe it is promising to design a more flexible mechanism to control the searching behaviors. For example, only the term-side inverted lists will be searched for some queries.

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

# Appendix

## A Baseline Details

• *Sparse Retrievers.* These methods represent documents and queries with sparse vectors over the vocabulary, then estimate relevance with overlapping entries. The retrieval operation is accelerated with the efficient inverted index (Zobel et al., 1998). **BM25** (Robertson and Zaragoza, 2009), the most basic sparse retriever. **DocT5** (Cheriton, 2019), extending BM25 by appending pseudo-queries to documents. **DeepCT** (Dai and Callan, 2019), learning contextualized term weights using BERT to replace the TF-IDF in BM25. **Uni-COIL** (Lin and Ma, 2021), learning contextualized term weights using BERT with contrastive learning. **DistilSPLADE** (Formal et al., 2021), learning sparse term-weight vectors over vocabulary with knowledge distillation.

• *Dense Retrievers with Brute Force Search (Flat).* These methods encode documents and

queries into dense embeddings, then estimate relevance with embedding similarity. For each input query, all document embeddings are evaluated. **DPR** (Karpukhin et al., 2020), the most basic dense retriever. **ANCE** (Xiong et al., 2021), enhancing DPR with hard negatives mined from the previous model snapshot. **CoCondenser** (Gao and Callan, 2022), retrieval-oriented pretraining the encoder model to compress more information in the embedding. **AR2** (Zhang et al., 2022), adversarially train the encoder and a ranker with knowledge distillation. RetroMAE (Xiao et al., 2022b), retrieval-oriented pretraining the encoder model with a shallow decoder and the representation bottleneck.

• *Dense Retrievers with Approximate Nearest Neighbor Indexes (ANNs).* These methods leverage ANN indexes to accelerate dense retrieval. **IVF-PQ** (Jégou et al., 2011a), the basic VQ index. **IVF-OPQ** (Ge et al., 2014), unsupervisedly learning a transformation orthogonal matrix for PQ to achieve higher accuracy. **IVF-JPQ** (Zhan et al., 2021a), optimizing the PQ codebook with contrastive learning towards retrieval quality. **Distill-VQ** (Xiao et al., 2022a), optimizing the IVF centroids and PQ codebook with knowledge distillation from any off-the-shelf embeddings. **HNSW** (Malkov and Yashunin, 2018), a powerful graph-based ANN index that is widely used in modern search engines (elasticsearch, 2015).

## B Implementation Details

For all methods involving clustering, we set the number of clusters $L$ to 10000 and the number of probing clusters when searching to 100 (except $HI^2$). For all methods involving PQ, we set the number of fragments $m$ to 96, the number of sub-clusters $k$ to 256, which results in 32 times smaller size than the full-precision one. For $HI^2$, we use the BERT's vocabulary (Devlin et al., 2019) as the term vocabulary $\mathcal{V}$, resulting in 30522 unique terms in total. $K_2^T$ is always set to 32 for both $HI^2_{unsup}$ and $HI^2_{sup}$.

For $HI^2_{unsup}$, we use KMeans over all document embeddings to produce cluster embeddings $\{e_{C_i}\}_{i=1}^L$, BM25 to produce term scores $s_v$ with $\alpha = 0.82, \beta = 0.68$, and OPQ (Ge et al., 2014) as the evaluation codec, all of which are unsupervised algorithms. $K^C$ is set to 25, $K_1^T$ is set to 15. For $HI^2_{sup}$, we initialize cluster embeddings with KMeans and optimize them afterwards. Note the cluster assignment $\phi(D)$ is fixed once initialized.

| Index | Codec | MS MARCO | | | NQ | | |
|---|---|---|---|---|---|---|---|
| | | R@100 | QL | IS | R@100 | QL | IS |
| $HI^2_{unsup}$ | default | 0.900 | 9 | 3.0 | 0.896 | 15 | 7.1 |
| | Flat | 0.909 | 18 | 28 | 0.900 | 31 | 65 |
| $HI^2_{sup}$ | default | 0.916 | 8 | 3.0 | 0.906 | 15 | 7.1 |
| | Flat | 0.920 | 17 | 28 | 0.907 | 31 | 65 |

Table 3: Evaluation of $HI^2$ with different codecs.

We use bert-base-uncased for the term selector. The passage is tokenized to 128 tokens before encoding. We employ the distilled OPQ (Xiao et al., 2022a) as the evaluation codec. $K^C$ is set to 30, $K_1^T$ is set to 3. For training $HI^2_{sup}$, we use the annotated ground truth document $D^+$, 7 hard negatives sampled from BM25 top 200 results, and in-batch negatives to form $D$.

In practice, we find the terms selected by $HI^2_{sup}$ results in much "denser" inverted lists than $HI^2_{unsup}$. In other words, some terms may be frequently selected from multiple passages, translating to their super big inverted lists. This is especially the case for Natual Questions. Therefore, on NQ, we prune the super big inverted lists to a moderate size inspired by static index pruning technique (Nguyen, 2009). Concretely, after indexing all documents, we count the size of each term-side inverted list, then take the one at the $\gamma$-th percentile ($\gamma$ defaults to 0.996) as the threshold, whereby inverted lists bigger than the threshold are identified as "super big". Next, we ascendingly order document references based on their individual score to the specific term of each super big inverted list. We prune the references from the head until the size of the inverted list equals the threshold.

## C Codec Analysis

Apart from the default PQ, $HI^2$ can be combined with other codecs. In Table 3, we compare PQ with the most powerful yet most expensive Flat codec. It can be observed that $HI^2_{unsup}$ and $HI^2_{sup}$ both benefit from the more powerful codec. This indicates that $HI^2$ can return high-quality candidates universally applicable for different codecs. It also reveals that the current PQ codec is still lossy. However, there is no free lunch: the powerful Flat codec comes with higher latency and higher index size, which is unfavorable for the index efficiency. In summary, we again verify the practicality of $HI^2$, one may flexibly balance between higher effectiveness and higher efficiency.