# OpenReview forum: "Hybrid Inverted Index Is a Robust Accelerator for Dense Retrieval"
_EMNLP/2023/Conference — EMNLP 2023 Main_

### Official Review · Reviewer_8PkW · 2023-07-31

**Soundness:** 4

**Excitement:**

3: Ambivalent: It has merits (e.g., it reports state-of-the-art results, the idea is nice), but there are key weaknesses (e.g., it describes incremental work), and it can significantly benefit from another round of revision. However, I won't object to accepting it if my co-reviewers champion it.

**Missing References:**

[1] MQH: Locality Sensitive Hashing on Multi-level Quantization Errors for Point-to-Hyperplane Distances. PVLDB 2022. https://www.vldb.org/pvldb/vol16/p864-lu.pdf

[2] Highly Efficient String Similarity Search and Join over Compressed Indexes. ICDE 2022. https://ieeexplore.ieee.org/document/9835221

[3] MILC: Inverted List Compression in Memory. VLDB 2017. http://www.vldb.org/pvldb/vol10/p853-wang.pdf

**Paper Topic And Main Contributions:**

This paper studied the problem of dense retrieval. It aims at improving the document embedding for clustering as well as accelerate the retrieval process. To this end, the authors proposed a hybrid inverted index which combines the advantage of dense retrieval and lexical matching. The authors also proposed learning techniques for term and cluster selection with a joint optimization solution. Experiments are conducted on several popular datasets and the results look promising.


**Questions For The Authors:**

What is the space overhead or memory consumption of propose techniques?

**Reasons To Accept:**

* The research topic is important.

* The paper is well written.

* The proposed techniques are solid.

**Reasons To Reject:**

* Some latest related works from the field of high dimensional similarity search are ignored such as [1]. They have been proved to have better performance than graph based methods such as HNSW compared here.

* The comparison with some inverted list based sparse retrieval methods are missing. There are some works about compression over inverted lists and perfrom similarity search on them such as [2] [3]. They could also be extended as baseline methods. Since they are optimized, they should perform better than the selected sparse method here.

* There is no result about space overhead.

**Reproducibility:**

4: Could mostly reproduce the results, but there may be some variation because of sample variance or minor variations in their interpretation of the protocol or method.

**Reviewer Confidence:**

2: Willing to defend my evaluation, but it is fairly likely that I missed some details, didn't understand some central points, or can't be sure about the novelty of the work.

---

> ### Author Rebuttal · Authors · 2023-08-29
>
> Dear Reviewer, we highly appreciate your recognization of the topic importance, the writing, and the technical depth of our work. We sincerely appreciate your input and value the opportunity to address your questions. We hope the following clarifications may address all your concerns and help to improve the current rating of our paper.
>
> > There is no result about space overhead. What is the space overhead or memory consumption of propose techniques?
>
> We report the memory consumption on MSMARCO when running ANN baselines and HI$^2$ (using their default settings and the Faiss toolkit) in the following table:
>
> |Method|Memory Consumption (GB)|
> |:-:|:-:|
> |IVF-PQ, IVF-OPQ, IVF-JPQ, Distill-VQ|2.6|
> |HNSW|30.5|
> |HI$^2$|4.1|
>
> It can be observed that the conventional IVF-PQ index family are the lightest. HI$^2$ consumes more memory because it needs to store the term-side inverted lists. However, since there are only a few terms selected from each document, the overhead is not so much. By comparison, HNSW consumes much more memory than IVF-PQ and HI$^2$, because it needs to store a big graph in the memory.
>
> > Some latest related works from the field of high dimensional similarity search are ignored such as [1]. They have been proved to have better performance than graph based methods such as HNSW compared here.
>
> We agree that some recent ANN indexes achieve better performance than HNSW, despite that HNSW is one of the most commonly used methods. However, we find that [a] is about LSH; according to the authority benchmarks on ANN (https://ann-benchmarks.com, https://github.com/erikbern/ann-benchmarks), HNSW is way more accurate than LSH under the same query throughput. Instead, we choose a well-known improvement of HNSW: NSG , for further experiment.
>
> |Method|Recall@100|Query Latency (ms)|Index Size (GB)|Memory Consumption (GB)|
> |:-:|:-:|:-:|:-:|:-:|
> |HNSW|0.887|6|28.0|30.5|
> |NSG|0.899|7|27.0|63.0|
> |HI$^2$|0.916|8|3.0|4.1|
>
> We may observe that using NSG indeed improves the retrieval quality. However, compared with HI$^2$, it has much bigger memory consumption, and inferior recall performance.
>
> It should be noted that although many recent ANN methods claimed improvements in different scenarios, HNSW and its close variations remain the dominant approaches and are widely referenced as competitive baselines in the area of information retrieval and semantic search [b,c,d]. They are also the most intensively used methods in libraries like Milvus and FAISS. Thus, the notable advantage over HNSW is indeed a strong indicator of HI$^2$'s value.
>
> [a] Fast Approximate Nearest Neighbor Search With The Navigating Spreading-out Graph. Fu, Cong, et al., Proceedings of the VLDB Endowment 12.5.
>
> [b] Jointly Optimizing Query Encoder and Product Quantization to Improve Retrieval Performance, Zhan et. al., CIKM'21
>
> [c] Learning Discrete Representations via Constrained Clustering for Effective and Efficient Dense Retrieval, Zhan et. al., WSDM'22
>
> [d] Constructing Tree-based Index for Efficient and Effective Dense Retrieval, Li et. al., SIGIR'23
>
> > The comparison with some inverted list based sparse retrieval methods are missing. There are some works about compression over inverted lists and perfrom similarity search on them such as [2] [3]. They could also be extended as baseline methods. Since they are optimized, they should perform better than the selected sparse method here.
>
> The listed references are about modification of inverted index [2] [3], such that the index can be more lightweight and efficient. Our method is not about modifying the inverted index, but unifying and optimizing the semantic and lexical features for the index, so that the retrieval quality can be improved. In other words, the listed references target on a parallel research direction with our work.
>
> The listed references are important works in their own field. If they can be extended and applied to common indexes on semantic search, like Lucene and FAISS, there will probably be an opportunity to make combination with HI$^2$ for further improvement.

---

### Official Review · Reviewer_wBPq · 2023-08-03

**Soundness:** 3

**Excitement:**

3: Ambivalent: It has merits (e.g., it reports state-of-the-art results, the idea is nice), but there are key weaknesses (e.g., it describes incremental work), and it can significantly benefit from another round of revision. However, I won't object to accepting it if my co-reviewers champion it.

**Paper Topic And Main Contributions:**

 This paper deals with the retrieval quality of dense retrieval and proposes a Hybrid Inverted Index that uses the embedding clusters and salient terms collaboratively to accelerate dense retrieval. It achieves lossless retrieval quality with competitive efficiency across a variety of index settings. The source code is provided.

**Reasons To Accept:**

1. The idea to combine embedding clusters and salient terms is interesting.
2. The paper is well organized. The description of the method design and implementation is clear and easy to follow.
3. The experimental results prove the effectiveness and efficiency of the proposed method on two popular benchmark datasets.

**Reasons To Reject:**

1. In the experiments, only the query latency is given. The setup latency should also be given.
2. The usage of “lossless retrieval quality” is easy to be confusing.

**Reproducibility:**

5: Could easily reproduce the results.

**Reviewer Confidence:**

3: Pretty sure, but there's a chance I missed something. Although I have a good feel for this area in general, I did not carefully check the paper's details, e.g., the math, experimental design, or novelty.

**Typos Grammar Style And Presentation Improvements:**

There are some typos such as “IVF-Flat and DistillIVF-Flat achieves recall” ([pdf](zotero://open-pdf/library/items/WJ33MJVQ?page=2)) , “2) Natural Questions (Kwiatkowski et al., 2019).”

---

> ### Author Rebuttal · Authors · 2023-08-29
>
> Dear Reviewer, thank you for your recognization of the idea, the writing, and the emperical performance of our work. We sincerely appreciate your input and value the opportunity to address your questions. We hope the following clarifications may address all your concerns and help to improve the current rating of our paper.
>
> > In the experiments, only the query latency is given. The setup latency should also be given.
>
> We evaluate the setup latency by measuring the time cost for constructing the index on MSMARCO. The results are reported in the following table:
>
> |Method|Setup Latency (minutes)|
> |:-:|:-:|
> |IVF-PQ, IVF-OPQ, IVF-JPQ, Distill-VQ|30|
> |HNSW|45|
> |HI$^2$|41|
>
> The setup of HI$^2$ takes a little more time than the conventional IVF-PQ/IVF-OPQ, because HI$^2$ needs to set up the inverted index for both lexical and semantic representations; however, the setup is faster than HNSW. It should be noted that the setup of index only needs to be conducted once in the offline stage. Therefore, the above difference won't be a bottleneck of HI$^2$'s application in practice.
>
> > The usage of “lossless retrieval quality” is easy to be confusing.
>
> We are sorry for the confusion caused for you. In our experiment, we treat the brute-force search of all documents (Flat) as the upper bound of retrieval quality. According to Table 1, HI$^2$ achieves Recall@100 of $0.916$ on MSMARCO and Recall@20 of $0.861$ on NQ, which is very close to the upper bound performance. We will think about a better term to describe this property.

---

### Official Review · Reviewer_4cBb · 2023-08-04

**Soundness:** 3

**Excitement:**

3: Ambivalent: It has merits (e.g., it reports state-of-the-art results, the idea is nice), but there are key weaknesses (e.g., it describes incremental work), and it can significantly benefit from another round of revision. However, I won't object to accepting it if my co-reviewers champion it.

**Paper Topic And Main Contributions:**

The manuscript describes a hybrid lexical and semantic inverted index for dense retrieval. The lexical component provides speed and precision due to terms; the semantic component allows to connect concepts beyond words. The semantic component works by clustering documents and creating meta-words that can be used to accumulate entries in the posting lists of the index. The approach includes a score function and a strategy to optimize lexical and semantic "words" selection.

**Questions For The Authors:**

A. I recommend measuring with brute force and the lexical+semantic score proposed, or some variation, to know how much precision/recall is missing due to the index approximation.

B. Please mention what kind of input receives each ANN and what kind of similarity/metric is used in each case.

C. Discuss the implications of using different representations for different ANN.

D. HNSW and IVFPQ are sensitive to their hyper-parameters. Did you optimize their hyper-parameters?

**Reasons To Accept:**

- The hybrid inverted index that it is fast and precise.
- It provides a way to combine lexical and semantic scores smoothly
- It introduces a way to distill the data to become smaller, and faster, while maintaining precise

**Reasons To Reject:**

Methodology weaknesses: It is not clear if they are seeing the same information as the hybrid index. If they are seeing the same, it is fair and it is a matter of speed the comparison, if they are seeing differences it is a matter of the score but speed is not necessarily comparable, at least as how the manuscript presents results as compared with IVFPQ and HNSW.


**Reproducibility:**

4: Could mostly reproduce the results, but there may be some variation because of sample variance or minor variations in their interpretation of the protocol or method.

**Reviewer Confidence:**

4: Quite sure. I tried to check the important points carefully. It's unlikely, though conceivable, that I missed something that should affect my ratings.

**Typos Grammar Style And Presentation Improvements:**

Table 1, Fig 3, Fig 4, Line 145. Natual -> Natural

---

> ### Author Rebuttal · Authors · 2023-08-29
>
> Dear Reviewer, we appreciate your recognization of the idea and the emperical performance of HI$^2$. We provide the following clarifications in response to your questions. Hopefully, these clarifications may address all your concerns and help to improve your rating of our paper.
>
> > Methodology weaknesses: It is not clear if they are seeing the same information as the hybrid index. If they are seeing the same, it is fair and it is a matter of speed the comparison, if they are seeing differences it is a matter of the score but speed is not necessarily comparable, at least as how the manuscript presents results as compared with IVFPQ and HNSW.
>
> All baselines see **the same information** as our method in the sense that all of them use the same text features for the queries and documents. However, they are differentiated in terms of how the representation is formulated on top of the text features. IVFPQ is based on the quantized semantic representation; HNSW is based on the semantic representation itself; our method (HI$^2$) is based on lexical representation and quantized semantic representation. IVFPQ (including all IVF-* and Distill-VQ), the sparse retrievers, and HI$^2$ are comparable in terms of speed, considering that all these methods are based on the inverted index. As for HNSW, the memory efficiency is comparable by measuring the index size. The running speed may not be directly comparable given that HI$^2$ and HNSW are based on different index structures. Their comparison is more of a reflection of the real-world performances when the two methods are applied (HNSW is based on the commonly used FAISS toolkit, whose implementation has been optimized and may serve as a proper indicator of its real-world performance).
>
> > A. I recommend measuring with brute force and the lexical+semantic score proposed, or some variation, to know how much precision/recall is missing due to the index approximation.
>
> The brute-force search, which enumerates every document embedding for each input query, is denoted as **Flat** in our paper. It serves as the upper bound for all ANNs (including HI$^2$) given the same embedding model because ANNs only selectively evaluate a small number of documents for each query. Besides, the hybrid retrieval (i.e. lexical+semantic score) is orthogonal to our method, because we unifies the lexical and semantic features at index level for accelerating dense retrieval, rather than interpolating their scores for higher effectiveness.
>
> We have reported the performance comparison with brute-force search in Table 1 in our paper, which is extracted and pasted below. It can be observed that HI$^2$ maintains a very little loss of precision (reflected by MRR) and recall compared with the upper bound. Particularly, HI$^2$ exhibits $-1.5\%$ absolute point loss in MRR and $-1.1\%$ absolute point loss in Recall with our default setting. However, it is indeed a tiny loss considering that there is more than x200 acceleration in terms of query latency.
>
> |Method|MRR@10|Recall@100|Query Latency (ms)|Index Size (GB)|
> |:-:|:-:|:-:|:-:|:-:|
> |Brute-Force Search (RetroMAE Flat)|0.416|0.927|1751|26|
> |HI$^2$|0.401|0.916|8|3.0|
>
>
> > B. Please mention what kind of input receives each ANN and what kind of similarity/metric is used in each case.
>
> All ANNs are based on **the same embedding models** (RetroMAE for MSMARCO and AR2 for NQ, which were the strongest methods on the corresponding datasets when our experiment was conducted).  The similarity is  measured by inner product, which follows exactly the original settings of RetroMAE and AR2.
>
> > C. Discuss the implications of using different representations for different ANN.
>
> In fact, all ANNs use **the same embedding model to produce semantic representations** (RetroMAE on MSMARCO and AR2 on NQ due to their highest brute-force search performance on the corresponding datasets when our experiment was conducted).
>
> For HNSW/IVF-PQ/IVF-OPQ, the ANN index is directly constructed based on the representations produced by the same embedding model. Therefore, there is no difference between their representations.
>
> For IVF-JPQ and Distill-VQ, the construction of their index calls for the adjustment of representations (i.e. the so-called joint optimization of index and embedding model). Thus, their representations are different from others. However, such a difference is introduced by the algorithms themselves, rather than our specification.
>
> > D. HNSW and IVFPQ are sensitive to their hyper-parameters. Did you optimize their hyper-parameters?
>
> We have comprehensively evaluated the performance of IVFPQ and HNSW (and other ANN baselines) with different hyper parameters (`nprobe` and `efSearch` respectively) and report the effectiveness-efficiency trade-off in **Figure 3**, where HI$^2$ consistently outperforms ANN baselines under different configurations. Part of the experiment results are cited here for analysis (please refer to the original figure for the complete result).
>
> |Method|nprobe|efSearch|$K^C$ (#Clusters to Search)|$K_1^T$ (#Terms to Index)|Recall@100|Query Latency (ms)|Index Size (GB)|
> |:-:|:-:|:-:|:-:|:-:|:-:|:-:|:-:|
> |IVFPQ|50|--|--|--|0.722|8|**0.9**|
> |IVFPQ|100|--|--|--|0.763|13|**0.9**|
> |IVFPQ|500|--|--|--|0.829|51|**0.9**|
> |HNSW|--|100|--|--|0.784|**2**|28|
> |HNSW|--|500|--|--|0.887|6|28|
> |HNSW|--|1000|--|--|0.905|18|28|
> |HI$^2$|--|--|10|1|0.881|**2**|3.0|
> |HI$^2$|--|--|30|3|0.916|8|3.0|
> |HI$^2$|--|--|60|6|**0.921**|14|3.0|
>
> Two critical facts can be observed:
>
> - HI$^2$ consistently and substantially outperform IVFPQ and HNSW with competitive or faster running speed. For example, given 2ms latency, HI$^2$ surpasses HSNW by $+9.7\%$ absolute point in Recall@100; given roughly 14ms latency, HI$^2$ surpasses IVFPQ by $+15.8\%$ absolute point in Recall@100.
> - The effectiveness of IVFPQ and HNSW benefit from larger `nprobe` and `efSearch`, respectively, yet with the cost of running speed. In HI$^2$, we can also adjust the hyper paremeters to achieve higher effectiveness with the cost of speed. It achieves the highest Recall@100 by doubling the number of clusters to search and the number of terms to index. Note that we do not need to re-build the inverted index even when we increase the number of terms to index due to our filtering-based implementation with Numpy ([anonymously open-sourced](https://anonymous.4open.science/r/HI2/src/utils/index.py)).

---

### Meta-Review · Area_Chair_bGbe · 2023-09-22

**Recommendation:** 4

**Metareview:**

The paper proposes a Hybrid Inverted Index that utilizes the embedding clusters and salient terms collaboratively to improve the quality and speed of dense retrieval.
All the reviewers have agreed that the paper has its merits in smoothly combining lexical and semantic features as an inverted index and bringing an instant speedup.
However, common concerns have also been raised by the reviewers that either the most recent baselines are missing or the speed reports are not very complete, making the results not that convincing.
During the rebuttal period, these concerns seem to be largely addressed.

---

### Decision · Program_Chairs · 2023-10-07

**Decision:**

Accept-Main

**Comment:**

The paper proposes a Hybrid Inverted Index that utilizes the embedding clusters and salient terms collaboratively to improve the quality and speed of dense retrieval.
All the reviewers have agreed that the paper has its merits in smoothly combining lexical and semantic features as an inverted index and bringing an instant speedup.
However, common concerns have also been raised by the reviewers that either the most recent baselines are missing or the speed reports are not very complete, making the results not that convincing.
During the rebuttal period, these concerns seem to be largely addressed.